# Single-Cell RNA Sequencing for Plant Research: Insights and Possible Benefits

**DOI:** 10.3390/ijms23094497

**Published:** 2022-04-19

**Authors:** George Bawa, Zhixin Liu, Xiaole Yu, Aizhi Qin, Xuwu Sun

**Affiliations:** 1State Key Laboratory of Crop Stress Adaptation and Improvement, School of Life Sciences, Henan University, 85 Minglun Street, Kaifeng 475001, China; ge.9410@yahoo.com (G.B.); lzx2021@henu.edu.cn (Z.L.); yxl86420@henu.edu.cn (X.Y.); qaz6835@henu.edu.cn (A.Q.); 2State Key Laboratory of Cotton Biology, School of Life Sciences, Henan University, 85 Minglun Street, Kaifeng 475001, China; 3Key Laboratory of Plant Stress Biology, School of Life Sciences, Henan University, 85 Minglun Street, Kaifeng 475001, China

**Keywords:** single-cell RNA-sequencing, cell-to-cell heterogeneity, transcriptomics, developmental trajectory, environmental stress adaptation

## Abstract

In recent years, advances in single-cell RNA sequencing (scRNA-seq) technologies have continued to change our views on biological systems by increasing the spatiotemporal resolution of our analysis to single-cell resolution. Application of scRNA-seq to plants enables the comprehensive characterization of both common and rare cell types and cell states, uncovering new cell types and revealing how cell types relate to each other spatially and developmentally. This review provides an overview of scRNA-seq methodologies, highlights the application of scRNA-seq in plant science, justifies why scRNA-seq is a master player of sequencing, and explains the role of single-cell transcriptomics technologies in environmental stress adaptation, alongside the challenges and prospects of single-cell transcriptomics. Collectively, we put forward a central role of single-cell sequencing in plant research.

## 1. Introduction

The use of sequencing technologies in plants to analyze genetic variation and metabolic regulation [1,2] has played a major role in enhancing our understanding of plant developmental processes and response to stimuli. However, the traditional sequencing method only generates average cell data and incapable of analyzing large number of cells, therefore losing cell heterogeneity information [3,4]. The technical reason behind this limitation is that the material or study sample used for traditional sequencing contains several cells that are mixed to obtain whole-genome sequence information of all cells [5]. However, the plant developmental process includes several regulatory factors and significant heterogeneity between different cells, which require a technology that enables cell heterogeneity and the discovery of new marker genes.

Moving from homogenous cells analysis to single-cell transcriptome requires reliable gene expression data from very low amounts of RNA present in a single cell [6]. Compared with traditional sequencing, single-cell RNA sequencing (scRNA-seq) technology, as the name implies, refers to the sequencing of a single genome for genomic or transcriptomic information that can reveal heterogeneity between cell populations. scRNA-seq methodologies have overcome the challenges of aggregate gene expression of whole tissue, enabling the identification of individual cells at high resolution, discovery of new cells, and better understanding of cell uniqueness [7,8], which has enabled scRNA-seq application in many plant species and enhanced our ability to compare and determine cell identity in plants. [9,10,11,12,13,14,15,16,17,18,19,20,21,22,23,24]. Additionally, scRNA-seq has facilitated our ability to gain new insights into possible results that were considered unattainable some years back, such as the generation of spatiotemporal gene expression atlas of heterogeneous shoot cells [25], first-generation gene expression map of the plant root [26], and the analysis of the molecular properties of thousands of cells in a single experiment [10,11]. The highly increased application of scRNA-seq in *Arabidopsis thaliana* [14,15,16,20,24,26,27,28,29,30,31] and other plants, such as *Zea mays* L. (maize) [13,18,19,32], *Oryza sativa* L. (rice) [17,33], *Nicotiana attenuata* (coyote tobacco) [22], *Solanum lycopersicum* (tomato) [25], and *Populus* L. (poplar) [34], suggests that single-cell sequencing has been accepted in plant research. These studies serve as key references for analyzing different mechanisms associated with plant development or responses to stimuli.

Here, we discuss recent advances in scRNA-seq approaches and the application of scRNA-seq in plant science, clarify why scRNA-seq sequencing is a top-notch method, and explain the role of single-cell sequencing technologies in environmental stress adaptation. Lastly, but most importantly, the challenges and possible prospects of scRNA-seq technologies are discussed.

## 2. The Power of Single-Cell Sequencing Methodologies

After the first scRNA-seq method was published [35], a large number of scRNA-seq methodologies have been proposed for scRNA-seq studies. scRNA-seq technologies have immense potential to reveal the mechanism of gene regulation and identification of cell-to-cell types and functions, which provide further insights into how developmental processes unfold within heterogeneous biological samples. The workflow for single-cell transcriptomics is summarized in Figure 1. With a global approach, somewhere in 2019, the Plant Cell Atlas (PCA) (https://www.plantcellatlass.org (accessed on 13 March 2022)) was established [36], with the objective of accumulating data for a broader understanding of the different plant types and combining high-resolution location information of nucleic acids, proteins, and metabolites within plant cells. The PCA uses scRNA-seq techniques to obtain genomic data from plant cells [36]. Plant and animal cells have certain things in common, especially their structures and roles. Plant cells are bigger than animal cells. Different plant species have different compositions and thicknesses according to the plant species, developmental level, specific tissue, and available environmental conditions [37]. These structural variations can possibly induce plant-specific, cell type-associated, and cell-positions-associated challenges in scRNA-seq analysis [14]. Therefore, the type of research design and planning is critical in scRNA-seq analysis in plant research.

Earlier technologies used to complete transcriptomic assays could only analyze hundreds of cells at very high resolution [38,39]. As a result of this limitation, these technologies have been replaced by high-throughput technologies, which can analyze thousands of cells at very high resolution. At the moment, the droplet-based method is commonly used for quantification purposes when dealing with large number of cells [40,41]. The remarkable expansion of scRNA-seq has been enabled by the development of droplet-based technology [40,41]. The droplet-based method can generate larger throughput of cells and lower sequencing costs per cell, and likewise the whole-transcript of scRNA-seq. The droplet-based method is best for accumulating large amounts of cells and identification of cell subpopulations of complex tissues, which makes this method dominant in the plant single-cell transcriptomics of *Arabidopsis* [14,15,16,18,26,27,28,42,43,44,45,46] and other plant species [17,22,33,47,48,49,50,51]. In addition, the application of the droplet-based technology into accessible commercial platforms has integrated this technology in modern plant research [13,14,16,18,26,27,28,33,42,49,50,51,52,53,54,55]. These studies prove how well single-cell transcriptomic can generate adequate information on cluster of cells according to their identity and response to stimuli. Aside from this commonly used droplet-based method is CEL-seq2, CEL-seq2 supports unique molecular identifier (UMI), and to lower amplification biases in CEL-seq2, mRNA amplification is often completed through in vitro transcription rather than PCR [48,56]. For the profiling of full-length RNA, the MARS-seq2.0 method, applied in a plate-based setup, has always been used [57]. This method provides variable information on the expression pattern of transcript isoforms [58] and detects abnormal expression of key genes. One limitation of the full-length sequencing technologies is the long processing time, which hinders its application in high-throughput single-cell analysis.

Despite the enormous contributions of these methodologies to making single-cell RNA sequencing the best approach for profiling rare or heterogeneous populations of cells, these technologies still have a number of challenges hampering the full functionality of this high-throughput sequencing approach. For example, the efficient isolation of individual cells, amplification of the genome, cost of querying the genome, long processing time, and interpretation of the data to reduce errors. Therefore, maximizing the quality of single-cell data and ensuring that the signals are separable from technical noise requires careful attention when designing single-cell experiments. However, the power of these technologies is only at an early stage, and considering scRNA-seq speed of expansion, we believe the future is yet to fully benefit from single-cell RNA transcriptomics.

## 3. Application of scRNA-seq in Plant Research

Recent advances in scRNA-seq approaches provide more opportunities for identifying cellular and molecular differentiation trajectory of plant stem cells at the single-cell level. The different applications of single-cell transcriptomics discussed below indicate the power of these technologies for redefining cell identities based on molecular analysis and for the identification of new cell differentiation routes. The first application of single-cell technologies is to uncover cell subtypes from heterogeneous cell populations. So far, the most profiled tissue by single-cell RNA sequencing is the *Arabidopsis* primary root tip [14,15,16,28,29,42,44,46,52,54,59,60]. For example, because a plant’s transcriptome changes within the day, Apelt and colleagues developed a high-resolution single-cell transcriptomic map of *Arabidopsis* root at the end of the day and above-ground tissues at the end of the day and end of the night and uncovered key markers for both time points. They found that, depending on the time of the day, single-cell transcriptome alterations occur in distinct tissues to a variant degree. Further analysis indicated that the most similar tissue type between root and above-ground tissue is dividing cells. From these observations, they investigated a previously uncharacterized marker of that cluster (MERCY1) and showed its function in meristematic development, demonstrating single-cell transcriptome role in identifying transcriptional heterogeneity for below- and above-ground tissue at specific time points [44]. Similarly, in *Arabidopsis* root tissue, Graeff et al. [61] applied scRNA-seq analysis to study the impact of brassinosteroid (BR) signaling in the root through characterization of briTRIPLE mutants at single-cell resolution. They found that BR signaling does not affect cell proliferation or cellular development, but rather promotes cell division plane orientation and cellular anisotropy. Single-cell sequencing profiling can identify phenotypic variations among cell types. To examine epidermal cells phenotypes, scRNA-seq was used to profile the mutants root hair deficient (rhd6) and glabrous2 (gl2) (non-hair cells). The data generated indicated the cell identity phenotypes. Interestingly, further transcriptional investigation in the abnormal epidermal cells in rhd6 and gl2 showed that hair cells in rhd6 were not completely changed to non-hair cells, and non-hair cells in gl2 were not completely changed to hair cells [26]. Emerging expansion of single-cell gene expression studies has enabled us to investigate transcriptional regulation in dynamic development processes and heterogeneous cell samples. In this study, gene expression of thousands of *Arabidopsis* root cells was investigated at the single-cell level. They found that root cells are heterogeneous, even within a single cell type. The pseudo time analysis showed that a single-cell root atlas reconstructed the continuous trajectory of root cell differentiation [29]. Single-cell RNA sequencing highly reconstructs cellular differentiation trajectories [13,14,15,26,27,28,42,43,46,48,49,50,52,55,59,62,63] Developmental trajectories have been successfully inferred from root single-cell transcriptome data [14,47,64,65,66,67]. Using single-cell transcriptomics, high-resolution profiling of *Arabidopsis* root was performed to create a plant cell atlas, which provided detailed information on developmental trajectories derived from the pseudo time analysis, which showed a finely resolved cascade of cell development from the niche through differentiation supported by high expression of interconnected genes [14].

In addition to the applications on *Arabidopsis* primary root tips, scRNA-seq has been applied to study *Arabidopsis* cotyledon development. For example, Liu and collaborators applied single-cell sequencing to analyze 5-day-old *Arabidopsis* cotyledons, identifying transcriptional networks regulating development from meristemoid mother cells (MMCs) to guard mother cells (GMCs) in the course of stomatal development [27]. More recently, the same team (Liu and collaborators) decided to apply single-cell analysis to uncover the mechanism underlying the early development of leaf veins in cotyledons. They found that the gene regulatory networks of some cell types showed potential roles of CYCLING DOF FACTOR 5 (CDF5) and REPRESSOR OF GA (RGA) in the early development and function of the leaf veins in cotyledons [24,68]. In brief, using single-cell sequencing, we can determine the cell type and function in any heterogeneous sample more accurately than before, which indicates a technology that is transforming the scope and depth of transcriptome analysis of cell populations.

Despite the increased application of single-cell sequencing in *Arabidopsis*, recent years have witnessed a rise in scRNA-seq technology applications in other crops, such as rice and maize, which could possibly be the tip of the iceberg considering the speed of growth of these technologies in plant research. For example, Liu et al. sequenced more than 20,000 single cells of rice root tip alongside computational analysis and in situ hybridization studies, which identified major cell types and specific marker genes. Using comparative analysis of single-cell expression data between rice genotypes and between rice and *Arabidopsis* enhanced the study of divergent characteristics of root cell type transcriptomics [33]. In addition, Zhang et al. used single-cell sequencing and chromatin accessibility to survey rice radicals. Through profiling of individual root tip cells, developmental trajectories of epidermal cells and ground tissues were reconstructed, which uncovered the mechanism regulating cell fate determination in these cell lineages. Further analysis uncovered transcriptome profiles and marker genes for these cell types [67]. Aside from rice, several studies have applied scRNA-seq toward crop improvement in maize by highlighting the transcriptional differentiation in maize cells at high resolution. For instance, Xu et al. (2021) recently used scRNA-seq technology to profile 12,525 single cells from developing maize ears. This profiling generated a scRNA-seq map of an inflorescence. They showed how the generated data could help promote maize genetics through possible identification of genetic redundancy, formation of gene regulatory networks at cell level, and identification of key loci with high ear yielding characteristics. Similarly, in the same year, Bezrutczyk et al. applied scRNA-seq to study bundle sheath (BS) differentiation in maize. The single-cell sequencing profiling helped uncover cells with unique characteristics on the adaxial side of the BS in maize, which could be essential for bioengineering of crops [50]. Even before the studies of Xu et al. [51] and Bezrutczyk et al. [50], scRNA-seq analysis was used to identify the landscape of cell states and the state of cell-fate acquisition in the developing of maize seedlings’ shoot apex, which opens the green light for future studies of maize development at single-cell resolution [49]. Aside from rice and maize, scRNA-seq has been applied to study other plants’ developmental processes or responses to stimuli [22,25,34]. The increased application of these technologies in several plant species suggests that future research will indeed bring single-cell transcriptomics to crop species and thus lead the way for its incorporation into applied plant research, which could benefit our agricultural systems in the future.

## 4. Why Consider Performing scRNA-seq in Plant Research?

Beyond the characterization of cells based on plant tissue anatomy, the rise of global transcriptomics analysis created an opportunity to group cell identity based on gene expression profiles [69,70,71,72]. However, bulk transcriptome experiments include several cell types. Until recently, microarray or bulk RNA-sequencing (bulk RNA-seq) has been used to perform plant gene profiling. Bulk RNA-sequencing provides targeted and bulk transcriptional data (Figure 2), thus restricting our understanding of the molecular mechanisms underlying the different regulatory processes [73]. Plants are made up of individual cells, which consist of unique characters, and performing bulk RNA-seq usually defeats such uniqueness and often fails to uncover new cell types (Figure 2). Thus, bulk RNA-seq analysis reflects the average transcriptional patterns across thousands of cells [3,4,74,75,76,77,78], which often does not provide heterogeneity information among individual cells. Earlier transcriptomics studies were usually applied to parts of tissues or whole plants. These conditions treated the material as a homogeneous sample, which averaged the differences of thousands of cells, thereby masking the uniqueness of each cell type. Shifting from broad-spectrum profiling of whole tissue to the state of more dedicated data sets containing one type of tissue has provided the needed resolution and identification of new cell types based on gene expression profiles [18,43,46,59]. So far, the advent of single-cell sequencing has provided unprecedented opportunities for profiling gene expression at the single-cell level. The main difference between single-cell sequencing and bulk RNA sequencing is the kind of data the two technologies generate.

In biological experiments, scRNA–seq data provide transcriptional heterogeneity among individual cells, while bulk RNA-seq techniques produce averaged gene expression information (Figure 2). Recent years have witnessed the rapid emergence of single-cell RNA–sequencing in plant science [14,16,17,22,23,25,26,27,28,29,33,46,49,51,52,61,67,79], and these studies have uncovered important cell-to-cell gene expression variability. Single-cell RNA sequencing technology enhances the comparison of transcriptomes of individual cells. scRNA-seq has been used to assess transcriptional similarities and variations within a population of cells, where previous reports have revealed high levels of heterogeneity in the root [11]. Therefore, heterogeneity analysis remains the main reason for conducting scRNA-seq analysis. The analysis of transcriptional differences has helped uncover rare cell populations, which could have gone undetected in pooled cells. Making cellular heterogeneity a priority, scRNA-seq can provide important information related to gene identification and function. For example, PHYTOCHROME INTERACTING FACTOR 4 (PIF4) and PIF5 expression was identified in meristemoid mother cells (MMCs) and guard cells (GCs) as key transcription factors that mediate the development of stomatal lineage cells in *Arabidopsis* [27]. In essence, scRNA-seq analysis of gene co-expression patterns could possibly identify co-regulated gene modules or gene-regulatory networks [27]. The removal of cells of interest from a whole tissue, which permits the identification of new cell types based on gene expression profiles, makes scRNA-seq a preferred technology compared to bulk RNA-seq analysis.

It is worth mentioning that, while scRNA-seq technology has been increasingly applied in plant science experiments, its successful functioning in any experiment depends on the methodology or protocol used and how successful the cells are isolated [80]. Cell isolation, which is the first step of scRNA-seq experiment, is the most challenging period, in which success depends on the kind of tissue and plant type. Ensuring a successful scRNA-seq experiment highly depends on the RNA quality produced for the library sequencing [81]. At the moment, enzymatic cell wall digestion and manual isolation are the two methods for cell isolation of single cells in plants. Although enzymatic cell wall digestion has been the most applied method for cell isolation for scRNA-seq studies, these two methods have some challenges when considered. Firstly, for enzymatic digestion, careful optimization and validation are needed in the biological and technical processes to recover the required cells from the whole tissue. Secondly, for manual cell isolation, this technique is tedious and associated with low throughput. However, despite these experimental challenges, the past years have seen a remarkable increase in single-cell transcriptomics not only in *Arabidopsis,* but also in other plants such as rice [17,33], maize [47,49,51], moss *P. patens* [62], and *Nicotiana attenuata* [22]. The tremendous increase in profiling gene expression at the single-cell level by these studies within this short period suggests a possible exponentially increase in single-cell transcriptomics in the coming few years, which sounds good for future plant research.

## 5. scRNA-seq for Responses to Biotic and Abiotic Stresses

Genetically alike cells growing in the same environment can change at the cellular level, indicating why some cells survive more severe stress than others. Cell-to-cell modifications in gene expression have been associated with various stress survival levels; however, the reason why transcript level changes across the transcriptome in a single cell is still emerging. Plant developmental stages or cell type-specific responses to the environment can be studied using single-cell transcriptomics approaches. As single-cell RNA sequencing continues to expand in plant research, recent years have seen a rise in gene profiling studies under different environmental conditions using scRNA-seq analysis. Abiotic stress stimuli change gene expression patterns in cell type-specific ways; however, for a given stress type, dissimilar stresses can induce transcriptional regulation of roughly the same set of genes. For example, a gene expression profile of about 120,000 single cells was isolated from *Arabidopsis* root to compare cellular growth of roots under sucrose or without sucrose, which induced some variations in cell type frequency and tissue-specific gene expression as a result of these external factors. Pseudo time analysis was used to study the transcriptional alterations during endodermis development, which revealed some important genes that function during the differentiation of this tissue, which show Arabidopsis root development at high resolution [28]. As part of a single-cell sequencing experiment using *Arabidopsis,* heat stress was applied to a whole seedling to help address any possibility of heterogeneity among cell types in response to abiotic stress. They found significant changes in expressions, such as cell type-specific expressions. It was observed that cells in the outer layer of the root had significant modifications in expression than the inner cell types [16]. In rice, using scRNA-seq, different major rice cell types were identified in response to abiotic stress, which revealed the heterogeneity among cell types of plants’ response to abiotic stress [17]. Macronutrients play an essential role in plant development, and any depletion in their levels negatively regulates plant growth [42]. It has been shown that plants can manipulate their growth behavior to survive unfavorable conditions [42]. Wendrich and colleagues profiled *Arabidopsis* root response to low phosphate conditions in soil using high-resolution single-cell transcript expression atlas of *Arabidopsis* root. They illustrated how plants increase their root hair density for better soil penetration for nutrient absorption. The cell data showed enrichment of specific gene function responses in root hair cells, which were related to increased biosynthesis of cytokinin in vascular cells at reduced phosphate levels, suggesting a possibility of cytokinin signaling in root hair responses to low phosphate levels in vascular cells [42]. Another body of work established cellular profiling of *Arabidopsis* leaves response to wounding to study de novo regeneration (DNRR). In the study, the transcriptional network was studied by detaching *Arabidopsis* leaves to study DNRR. The single-cell profiling data showed gene expression patterns in the wounded area of detached leaves during adventitious rooting [31]. By analyzing the response of different cell clusters to different stress treatments, scRNA-seq can uncover cellular activities that are crucial to plant growth and adaptation to adversity while also revealing the balance of plant growth and resilience. Importantly, these studies demonstrated a possible heterogeneity among cell types of plants’ responses to abiotic stress using scRNA-seq approaches.

scRNA-seq is revolutionizing the study of phenotypic cell-to-cell variation in eukaryotes, but with technical hurdles in exploring the analysis of cell-to-cell variation in prokaryotes [82]. Bacteria respond to environmental changes with specific transcriptional networks; however, within the same genetic populations, these programs are not homogenously expressed [83]. Using an advanced poly(A)-independent scRNA-seq approach, Imdahl and colleagues reported the expression of growth-dependent genes in individual Salmonella and Pseudomonas bacteria across all RNA classes and genomic regions [82], providing a benchmark data resource for scRNA-seq of other bacteria species and host–pathogen interactions. Again, in bacteria, individual cells within an isogenic population show several stress response mechanisms; however, the nature of this heterogeneity is not clear. Using scRNA-seq approaches, cell-to-cell heterogeneity in yeast transcriptome was profiled. The study demonstrated changes in transcription factors, which suggested that cells change their transcriptome with or without stress conditions. scRNA-seq profiling and live-cell imaging of transcription factor activation analysis showed cells with high-stress responses and cells with low-stress responses in a study by Gasch et al. [84]. These studies highlight the transcriptional heterogeneous nature of single-cell transcriptomics in biotic stress response. Nevertheless, we believe that the current contribution of single-cell RNA-sequencing in plant biotic stress responses is just the tip of the iceberg since recent scRNA-seq studies are beginning to merge with spatial transcriptome (ST), as well as mass spectrometry imaging (MSI)-based spatial metabolome (SM), to systematically study the spatial dynamics of transcriptome and metabolome in different tissues and cell types during plant responses to stress.

## 6. Conclusions and Perspectives

Single-cell sequencing as a technology for characterizing the state of a cell across multiple molecular layers has been shown to be powerful beyond imagination in recent times. Recent years have witnessed a surge in the number of related scRNA-seq publications, suggesting the acceptance of this technology in plant science. This review discusses the power of single-cell transcriptome technologies, the application of single-cell sequencing, the major role of single-cell sequencing methods in cell heterogeneity, the role of single-cell sequencing technologies in environmental stress adaptation, and the limitations and prospects of single-cell transcriptomics. Despite the increased expansion of single-cell transcriptome in plant science, we believe the future is yet to see how this technology will unprecedentedly uncover interesting insights into the regulatory mechanisms of cell identity in crop science, which implies that, in the not-so-distant future, single-cell sequencing should be increasingly applied to several non-model crops to develop more productive plant varieties to help increase and expand crop production.

Nevertheless, despite the extensive applicability, single-cell sequencing technology at single-cell resolution does not completely explore the complex developmental process of plant development or response to stimuli, such as the inability to provide cellular spatial information in single-cell RNA sequencing. This is because, considering scRNA-seq alone, the integrative, multiomics analysis of single cells, such as with spatial transcriptome and mass spectrometry imaging-based spatial metabolome, provides spatial information of cells at single-cell resolution [11]. In addition, emerging evidence has shown that transcriptomic profiling of complex tissues by single-nucleus RNA-sequencing (snRNA-seq) has some advantages over scRNA-seq [25]. For instance, the existence of plant cell walls significantly hampers scRNA-seq in plant research. Protoplasting has been applied for scRNA-seq analysis but mostly limited to model plants, such as *Arabidopsis*, which has a stable and well-developed protoplasting protocol. Meanwhile, several cell types are resistant to protoplasting, and protoplasting may promote ectopic gene expression and deletion of certain cell types, which increases the bias proportion of cell types. To overcome this protoplasting limitation, Tian et al. established a plant tissue processing pipeline to isolate high-quality nuclei, which are compatible with high-throughput snRNA-seq [25]. Aside from Tian et al. studies, Wang and colleagues recently developed an efficient protocol for protoplast preparation in Chirita pumila, consisting of two digestion processes with several enzymatic buffers. This protocol generates viable cell suspensions suitable for scRNA-seq analysis [85]. Moreover, single-cell sequencing transcriptomics at the moment cannot capture information for both eukaryotic and prokaryotic organisms at the same time because current technologies are limited to eukaryotic cells having mRNA polyadenylation. At the moment, single-cell sequencing is facing these temporal challenges because joint studies on plants’ spatial and single-cell levels are rare, although this combination has been advanced in the human and animal fields [86,87]; it is time to construct a spatial and single-cell resolution map for single-cell studies in plants, because such integration provides the cellular spatial information in single-cell RNA sequencing and could help solve the challenges of how to quantify the transcript abundance of both plant and prokaryotic cells in a symbiotic form while preserving their spatial context, which means that the integration of single-cell sequencing, spatial transcriptome, and metabolome analysis is expected to revolutionize plant science research at an unprecedented level beyond what we are seeing now. Hence, in the future, we hope to see single-cell sequencing integrated with these technologies for throughput expansion of single-cell sequencing, which we believe will provide breakthroughs in the field of single-cell transcriptomics.

## Figures and Tables

**Figure 1 ijms-23-04497-f001:**
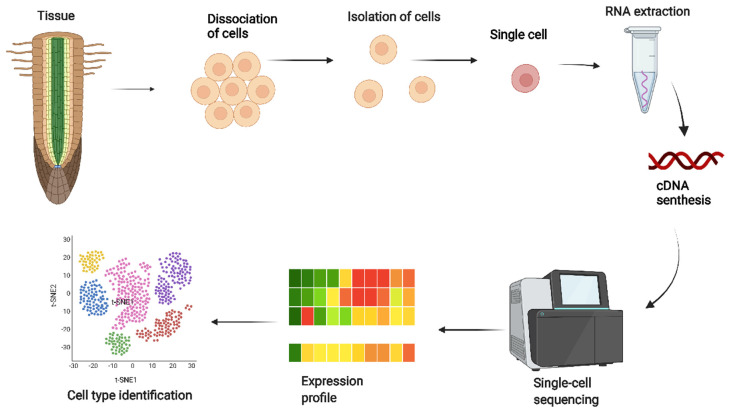
Proposed workflow for single-cell RNA sequencing. Starting from dissociating target cells from the tissue, cell isolation, RNA extraction, cDNA synthesis by reverse transcriptase, single-cell sequencing, expression profile, and cell-type identification. Single-cell RNA sequencing enables unbiased, high-throughput, and high-resolution transcriptomic analysis of individual cells in plants.

**Figure 2 ijms-23-04497-f002:**
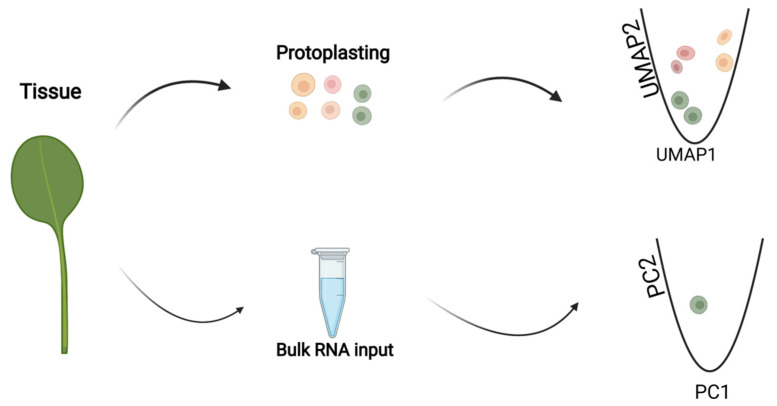
Proposed model illustrating the difference between bulk RNA sequencing and single-cell RNA sequencing. In biological samples, single-cell transcriptomic data provide transcriptional heterogeneity among individual cells, while bulk RNA-sequencing techniques produce average gene expression information.

## Data Availability

This study did not generate any unique datasets or code.

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
