# Peer review of "Single-Cell RNA Sequencing for Plant Research: Insights and Possible Benefits"

_ijms, 2022, doi:10.3390/ijms23094497_

Round 1

Reviewer 1 Report

In this manuscript, Bawa et al., review the recent reports on single-cell RNA sequencing (scRNA-seq) in plant biology. They introduce the methods and application of the scRNA-seq and discuss the challenges and prospects of scRNA-seq. Given that the scRNA-seq has emerged as a kind of paradigm-shifting technology in the genomics field, this manuscript is a timely review and will be beneficial for the plant science community. The followings are my comments on this manuscript:

  1. Efforts to overcome the limitation of the current scRNA-seq: As pointed out in the manuscript, there are several key limitations of the current scRNA-seq, such as protoplast isolation and diverse cell size in plant tissues. To overcome this limitation, several methods were proposed. For example, nuclei isolation rather than protoplasting is widely utilized. It will be informative if the authors can introduce recent methods in this review.
  2. The examples of scRNA-seq in other plants: Although most of the scRNA-seq studies were reported in Arabidopsis, it will be beneficial if the authors can introduce other plant studies, such as tomato, poplar, and other non-model plants.
  3. Long paragraph: This is a bit of technical advice to help the readers. Each section was written in just one or two paragraphs. It is difficult to read the different stories in a long paragraph. Please consider separating it into several paragraphs.
  4. Authors: On line136, 190, 206, 208, all the authors were listed. Please indicate them as XXX et al., instead.
  5. Reference 80: This study was not conducted using scRNA-seq. They used the cell-sorting method. Please clarify this in the manuscript.

Reviewer 2 Report

Here, Bawa and colleagues present a critical review on perspectives of single-cell RNA sequencing (scRNA-seq) for plant research. The article is timely and logically written; authors clearly articulate that scRNA-seq is of utmost importance in contemporary plant research and might lead to the significant advances in developmental and functional biology of the plants. As to my eye, I cannot see any glaring omissions and would recommend acceptance of the paper as is.

The authors first state the definition of scRNA-seq and differentiate it from bulk RNA sequencing, clearly postulating the advantages of scRNA-seq, first and foremost conceptually another level of spatiotemporal gene expression resolution permitting the identification of subtle cellular phenotypes. In the next section, authors discuss the historical development of scRNA-seq technology including technical pitfalls such as technical difficulties in individual cell isolation (e.g. the proportion of dead cells, doublets and empty droplets in the cell suspension), high cost of amplification and quering the genome, quite long processing time and complexity of the bioinformatic analysis. Here, authors conclude that despite scRNA-seq is now extensively applied for human and animal research, this field is still in its infancy for plant research and is yet to fully benefit from single-cell RNA transcriptomics.

Further, authors supply the readers with a number of examples of using scRNA-seq in plant research (e.g. for tracking the transition from meristemoid mother cells to guard mother cells in the course of stomatal development) and talk over the rationale behind expansion of scRNA-seq in plant research, for instance, to assess a response to biotic and abiotic stresses.

Further recommendations for the authors (that can be stated in Conclusions and Perspectives Section if authors wish) might include the focused discussion on applying scRNA-seq technology in farming, crop science and crop production (currently, this is sparsely articulated throughout the article). To me, the biggest advantage we can gain from scRNA-seq in plant research is its use to develop novel, more productive, plant varieties that might be able to overcome the upcoming food shortage worldwide. From this side, authors can include the discussion on cost-efficiency of expanding scRNA-seq technologies from the research institutions and molecular/cell biology industry into crop production facilities. Authors might also refer to other papers on scRNA-seq which previously focused on challenges in scRNA-seq in human and animal setting, e.g. bioinformatic tools for unbiased interrogation of RNA-seq data, particularly resolution of cluster identification. In any case, the paper is good and deserves publication in the International Journal of Molecular Sciences.
